# Analytical Validation of the Cxbladder^®^ Triage Plus Assay for Risk Stratification of Hematuria Patients for Urothelial Carcinoma

**DOI:** 10.3390/diagnostics15141739

**Published:** 2025-07-08

**Authors:** Justin C. Harvey, David Fletcher, Charles W. Ellen, Megan Colonval, Jody A. Hazlett, Xin Zhou, Jordan M. Newell

**Affiliations:** 1Pacific Edge Diagnostics NZ, Ltd., 87 St. David Street, Dunedin 9016, New Zealand; charles.ellen@pacificedge.co.nz (C.W.E.); megan.colonval@pacificedge.co.nz (M.C.); jody.hazlett@pacificedge.co.nz (J.A.H.); xin.zhou@pacificedge.co.nz (X.Z.); 2David Fletcher Consulting Ltd., 67 Stornoway Street, Karitane 9471, New Zealand; david@davidfletcher.consulting; 3Pacific Edge Diagnostics USA, Ltd., 1214 Research Boulevard, Hummelstown, PA 17036, USA; jordan.newell@pacificedgedx.com

**Keywords:** analytical validation, biomarkers, droplet-digital PCR, hematuria, quantitative RT-PCR, urothelial carcinoma, urinary tests

## Abstract

**Background/Objectives**: Cxbladder^®^ Triage Plus is a multimodal urinary biomarker assay that combines reverse transcription-quantitative analysis of five mRNA targets and droplet-digital polymerase chain reaction (ddPCR) analysis of six DNA single-nucleotide variants (SNVs) from two genes (fibroblast growth factor receptor 3 (*FGFR3*) and telomerase reverse transcriptase (*TERT*)) to provide risk stratification for urothelial carcinoma (UC) in patients with hematuria. This study evaluated the analytical validity of Triage Plus. **Methods**: The development dataset used urine samples from patients with microhematuria or gross hematuria that were previously stabilized with Cxbladder solution. Triage Plus was evaluated for predicted performance, analytical criteria (linearity, sensitivity, specificity, accuracy, and precision), extraction efficiency, and inter-laboratory reproducibility. **Results**: The development dataset included 987 hematuria samples. Compared with cystoscopy (standard of care), Triage Plus had a predicted sensitivity of 93.6%, specificity of 90.8%, positive predictive value (PPV) of 46.5%, negative predictive value of 99.4%, and test-negative rate of 84.1% (score threshold 0.15); the PPV increased to 74.6% for the 0.54 score threshold. For the individual *FGFR3* and *TERT* SNVs, the limit of detection (analytical sensitivity) was a mutant-to-wild type DNA ratio of 1:440–1:1250 copies/mL. Intra- and inter-assay variance was low, while extraction efficiency was high. All other pre-specified analytical criteria (linearity, specificity, and accuracy) were met. Triage Plus showed good reproducibility (87.9% concordance between laboratories). **Conclusions**: Cxbladder Triage Plus accurately and reproducibly detected *FGFR3* and *TERT* SNVs and, in combination with mRNA expression, provides a non-invasive, highly sensitive, and reproducible tool that aids in risk stratification of patients with hematuria.

## 1. Introduction

The American Urological Association (AUA) microhematuria guideline recommends risk stratification of patients with microhematuria for the presence or absence of urothelial carcinoma (UC) [1]. Based on the 2025 AUA risk criteria, clinicians should engage low-risk patients in repeat urinalysis within 6 months, while intermediate- and high-risk patients should undergo full work-up (i.e., cystoscopy and renal ultrasound in intermediate-risk patients, or cystoscopy and axial upper tract imaging in high-risk patients). According to the AUA 2025 guidelines, validated urinary biomarker tests or cytology may be used to guide shared decision making for deferred work-up in intermediate-risk patients who are reluctant to undergo full work-up, with repeat urinalysis conducted within 1 year [1].

Cystoscopy, while considered the gold standard for UC diagnosis, may produce false-negative and false-positive results and has variable performance [2]. Historically, cytology and fluorescence in situ hybridization have also been used to evaluate patients with suspected UC; however, both of these methods have poor sensitivity [3,4]. Urine cytology may be used to provide clinical resolution in samples with equivocal cystoscopy findings, but it may miss some high-grade or muscle-invasive tumors due to low sensitivity [5]. Therefore, there is an unmet need for non-invasive tests with high sensitivity and specificity to assess the risk of UC in patients with hematuria.

Cxbladder^®^ Triage and Detect assays are validated reverse transcription quantitative-polymerase chain reaction (PCR)-based urinary biomarker tests that are used for risk stratification of patients with hematuria [6,7]. Both assays quantify mRNA expression of five biomarkers, including four genes that are known to be associated with UC (cyclin-dependent kinase 1 (*CDK1*), midkine (*MDK*), insulin-like growth factor binding protein 5 (*IGFBP5*), and homeobox A13 (*HOXA13*)) [8] and one known marker of inflammation (C-X-C motif chemokine receptor 2 (*CXCR2*)) [6,7]. Cxbladder Triage also incorporates clinical risk factors to increase the assay sensitivity (age, sex, smoking history, and presence of frequent gross hematuria) [9].

The multimodal Cxbladder Triage Plus assay was subsequently developed with ongoing research. It combines the five mRNA biomarkers with six DNA single-nucleotide variants (SNVs) from two other genes that are known to be associated with UC: fibroblast growth factor receptor 3 (*FGFR3*) and telomerase reverse transcriptase (*TERT*) [10]. Hotspot mutations in the *FGFR3* gene (R248C, S249F/C, G372C, and Y375C (missense gain-of-function mutations)) and upstream of the *TERT* promoter (C228T and C250T (gain-of-function mutations)) are often identified in urothelial carcinoma samples [11,12,13]. In a previous clinical validation study, an enhanced Triage assay showed improved diagnostic performance over the original Detect and Triage assays [14]. Of note, an earlier version of Triage Plus was named “Detect+” in the Lotan et al. 2023 publication [14]; this should not be confused with “Triage+”, which contained clinical factors. Further refinement of the enhanced Triage assay resulted in development of the Cxbladder Triage Plus assay (hereafter referred to as Triage Plus), which provides higher specificity and sensitivity than earlier versions of the assays while solely relying on genomic biomarkers (i.e., without the need for clinical factors).

For biomarker tests to be integrated into routine use in clinical practice, they must show strong analytical validity, clinical validity, and clinical utility [15]. A previous analytical validation study showed accurate and reproducible quantification of mRNA expression for the five biomarker genes with both Cxbladder Triage and Detect assays [16]; as Triage Plus also uses these biomarkers, the analytical performance of mRNA detection was not changed and so will only be referred to here (not reiterated). This analytical validation study focused on the linearity, analytical sensitivity, specificity, accuracy, and precision, as well as the extraction efficiency and inter-laboratory reproducibility, for the analysis of the six DNA SNVs in Triage Plus, and we reference the earlier publication [16] for the mRNA analysis.

## 2. Materials and Methods

### 2.1. Samples and Study Design

A development dataset was used to develop the Triage Plus algorithm and to perform the analytical validation. The dataset included −80 °C stored Cxbladder-stabilized urine samples from patients with gross hematuria or microhematuria who had participated in a previous clinical validation study [14] and from patients with hematuria who participated in the STRATA: Safe Testing of Risk for Asymptomatic Microhematuria study [17]. Samples were blinded prior to their use in the development dataset.

The previous clinical validation study included two cohorts of patients aged ≥18 years with gross hematuria (United States (US)) or aged >21 years with gross hematuria or microhematuria (Singapore) who were scheduled to undergo evaluation for possible UC [14]. The study protocols were approved in Singapore by the SingHealth Centralized Institutional Review Board (IRB; 2016/2471 (approved 19 July 2016, 13 June 2017, and 23 February 2019)) and NHG Domain Specific Review Board (2018/00234-SRF0002 (approved 3 August 2019)), and in the US by local relevant IRBs (Chesapeake IRB, Pro00009623 (approved 13 December 2012, 29 January 2013, 6 February 2013, 3 June 2014, 24 February 2016, 9 May 2016, and 20 May 2016); Florida Hospital IRB, 394399 (approved 19 March 2013); UT Southwestern Medical Center IRB, STU-112012-018 (approved 4 April 2013); and PennState Hershey IRB, 41719EP (approved 5 June 2013)). These studies were conducted in accordance with the Good Clinical Practice requirements and the Declaration of Helsinki (1975, revised 2013), and all patients provided informed consent before any study procedures were undertaken [14].

The STRATA study was a multicenter US study of patients aged > 18 years who were referred for evaluation of microhematuria [17]. Patients with a prior history of urologic malignancy or pelvic radiotherapy were excluded. The study protocol was approved by the PennState IRB (STUDY00010988 (approved 9 September 2019)), the University of Southern California IRB (HS-19-00766 (approved 14 June 2020)), the UT Southwestern Medical Center IRB (STU-2019-1020 (approved 11 September 2019)), the University of British Columbia Clinical Research Ethics Board (H19-01797 (approved 7 November 2019)), the University of Minnesota IRB (STUDY00008103 (approved 27 April 2020)), the Vanderbilt IRB (200304 (approved 16 June 2020)), the Western Research Health Sciences Research Ethics Board (114112 (approved 3 February 2020)), and the WCG IRB (20202112 (approved 20 July 2020, 2 August 2021, 21 November 2021, 19 October 2022, and 13 March 2023)). This study was also conducted in accordance with the Good Clinical Practice requirements and the Declaration of Helsinki (1975, revised 2013), and all eligible patients signed IRB-approved consent prior to study entry [17].

### 2.2. Algorithm Development

In the original Detect+ algorithm, quantification of mRNA expression was encapsulated by two variables in the algorithm: *X*_1_, which combined the four biomarkers known to be associated with UC (*IGFBP5*, *HOXA13*, *MDK*, and *CDK1*) into a single predictor; and *X*_2_, the inflammation target (*CXCR2*). The DNA component of the original Detect+ algorithm was summarized as an “*FGFR3* and *TERT* DNA-positive call” or a DNA-positive result.

Based on a thorough review of the statistical and machine-learning literature, which was conducted to select potentially useful algorithms, it was concluded that the best predictions were obtained using a single algorithm (i.e., Bayesian additive regression tree (BART)) [18]. However, the BART algorithm could not improve the performance of the original Detect+ algorithm.

After re-evaluation of this approach, while keeping the *FGFR3* and *TERT* DNA-positive call, the mRNA quantification component of the Detect+ algorithm was improved to create the new algorithm, hereafter referred to as the Triage Plus algorithm.

In the Triage Plus algorithm, all five biomarkers appeared separately in the second-order polynomial equation (*X*_1_ = *IGFBP5*, *X*_2_ = *HOXA13*, *X*_3_ = *MDK*, *X*_4_ = *CDK1*, and *X*_5_ = *CXCR2*), as follows:$$logit\left(p\right)=a_{0}+a_{1}X_{1}+a_{2}X_{2}+a_{3}X_{3}+a_{4}X_{4}+a_{5}X_{5}+a_{11}{X_{1}}^{2}+a_{22}{X_{2}}^{2}+a_{33}{X_{3}}^{2}\;\;+a_{44}{X_{4}}^{2}+a_{55}{X_{5}}^{2}+a_{12}X_{1}X_{2}+a_{13}X_{1}X_{3}+a_{14}X_{1}X_{4}+a_{15}X_{1}X_{5}\;\;+a_{23}X_{2}X_{3}+a_{24}X_{2}X_{4}+a_{25}X_{2}X_{5}+a_{34}X_{3}X_{4}+a_{35}X_{3}X_{5}+a_{45}X_{4}X_{5}$$

The coefficients (*a*_0_, *a*_1_, …, *a*_45_) were obtained by fitting a logistic regression model, with the confirmed diagnosis as the response variable and the linear predictor as given by the equation above. The Triage Plus score, which was given by the value of *logit* (*p*) determined by the above equation using the inverse of the *logit* function, can be interpreted as an estimate of the probability of cancer. The calculated composite Cxbladder Triage Plus score ranged from 0.00 to 1.00. Two test thresholds were set, with one threshold optimized for maximum sensitivity and negative predictive value (NPV) and the second threshold chosen to maximize positive predictive value (PPV), to create three zones to risk-stratify patients on the probability of having UC: low, intermediate, or high probability.

The development dataset was used to determine the thresholds for the predicted performance of Triage Plus and obtained thresholds of 0.15 and 0.54. Using these thresholds and compared with cystoscopy, the predicated performance parameters were predicted sensitivity, specificity, PPV, NPV, and test-negative rate (TNR). All performance outcomes were based on “leave one out” cross validation (i.e., the score for each sample was found by training the relevant algorithm on all other samples).

### 2.3. Analytical Validation

Assessment of the analytical performance of Triage Plus for detection of RNA from *IGFBP5*, *HOXA13*, *MDK*, *CDK1*, and *CXCR2* genes was conducted as previously described for the Cxbladder Triage and Detect assays [16]. The methods for the analytical validation of Triage Plus for detection of the six DNA SNVs from *FGFR3* (R248C, S249F/C, G372C, Y375C) and *TERT* (C228T, C250T) are described in the following sections.

#### 2.3.1. Linearity

The linearity of Triage Plus was assessed for each of the six DNA SNVs of *FGFR3* and *TERT*, with an upper limit being 2.21 λ and 1.76 λ, respectively (where λ is the mean number of target DNA molecules per droplet in droplet-digital PCR (ddPCR) using Poisson distribution). A 10-point dilution curve (33,000, 6600, 1320, 264, 132, 66, 33, 16.5, 8.25, and 4.12 DNA copies/well, with at least four replicates at each concentration) was performed to confirm the linearity of each analytic target. Statistical testing was performed to determine the concentration at which the assay became non-linear for each analytic target. Data were fitted using a linear regression model, and linearity was assessed using the regression coefficient (*R*^2^) and mean squared error (MSE). The null hypothesis was that Triage Plus had valid linear regression through the whole diluted range; this null hypothesis was rejected if *R*^2^ was <0.9. The MSE was used to compare linearity between different analytic targets.

#### 2.3.2. Analytical Sensitivity

Analytical sensitivity (or limit of detection (LOD)) was defined as the lowest analyte concentration that could be consistently detected with 95% probability. The LOD for *FGFR3* and *TERT* DNA SNVs was determined by logistic regression and the three-concentrations approach. For logistic regression, a model was fitted to explain the relationship between the dependent (positive/negative) and independent (analyte concentration) variables, and the LOD was predicted based on this fitted logistic model. For the three-concentrations approach, a fraction of positives was calculated from highest to lowest concentration, and the concentration that matched 95% of positives ($c_{0}$) was the boundary of LOD. Samples with concentrations $>c_{0}$ were grouped into three concentration levels (i.e., $c_{1}$, $c_{2}$, and $c_{3}$, where $c_{1}>c_{2}>c_{3}>c_{0}$). If $n_{i}$ was the number of positive samples at concentration $c_{i}$, it was assumed that $n_{i}\sim Poisson({\lambda c}_{i})$ (where *i* = 1, 2, 3); the expected value $E\left(n_{i}\right)={\lambda c}_{i}$ was the mean number of positive samples at concentration $c_{i}$. If $p_{i}$ was the probability of no positive samples at concentration $c_{i}$, then $p_{i}=P\left(n_{i}=0\right)=e^{-\lambda ci}$. If $c^{*}$ was the LOD, by definition, $1-p_{i}=0.95$ and $p_{i}=e^{-\lambda c*}=0.05$; hence, $c^{*}=-\log0.05/\hat{\lambda }$. If $X_{i}$ was the observed number of positive samples at concentration $c_{i}$, the maximum likelihood estimator for λ ($\hat{\lambda }$) could be computed using the Triage Plus algorithm. The three concentrations ($c_{1}$, $c_{2}$, and $c_{3}$) could be selected in multiple ways and the corresponding number of samples at each concentration changed accordingly, with both having an impact on the LOD estimate. The most conservative value was used as the estimated LOD.

#### 2.3.3. Analytical Specificity

Analytical specificity was defined as the ability of Triage Plus to detect *FGFR3* and *TERT* mutant DNA SNVs in the presence of potentially interfering substances, which may have been carried over from the extraction reagent or present in the patient’s urine sample. The effect on analytical specificity of the assay for both sample- and process-derived interfering substances was assessed.

The sample-derived substances assessed were red blood cells (RBCs; 8 × 10^5^, 4 × 10^6^, 2 × 10^7^, and 1 × 10^8^ cells/mL), bacteria (1 × 10^6^ cells/mL *Escherichia coli*), yeast (1 × 10^4^ colony-forming units (CFU)/mL), urea (60 mg/mL), glucose (0.5 mg/mL), and protein (1.25, 2.50, 5.00, and 10.00 mg/mL serum albumin). The selected amount of each substance was mixed with eight high- and low-concentration *FGFR3* and *TERT* extraction controls. The contaminated controls were then extracted and compared with high- and low-concentration controls that did not contain interfering substances. Samples within the expected level of gene variance and with a Triage Plus score that was within the 95% confidence interval (CI) for the control were considered acceptable.

The process-derived substances assessed were ethanol (4%), MagMAX wash buffer (2%), Cxbladder stabilizing reagent (1%), MagMAX magnetic beads (5%), and acetone (10%). The substance percentage was calculated per 64 μL of elution volume and each substance was mixed with high and low concentrations of *FGFR3* and *TERT* extraction controls as 12 replicates and compared with controls (without potentially interfering substances) to determine if the samples were affected by the process-derived contaminants.

#### 2.3.4. Analytical Accuracy

As there was no analytical standard for *FGFR3* and *TERT* DNA, multiplex ddPCR was considered the most accurate method to define absolute quantitation in Triage Plus. To validate the analytical accuracy of Triage Plus, ddPCR of the SNVs of *FGFR3* (R248C, S249F/C, G372C, and Y375C) and *TERT* (C228T, C250T) as single analytes were compared with combined analyte samples (of mutant plus wild type (WT) DNA). A combination of *TERT* C228T and C250T mutant DNA was also assessed.

Mutant DNA alone and mutant DNA combined with WT DNA were manufactured in parallel to contain equivalent concentrations of the target DNA. High-extraction controls (HECs) had a DNA concentration of ~1 × 10^6^ copies/μL and low-extraction controls (LECs) had a DNA concentration of ~1 × 10^4^ copies/μL. Each mutant was combined with WT DNA at a high (1:10) and low (1:200) mutant-to-WT ratio for the HECs and LECs, respectively. Mutant DNA samples were compared with mutant + WT DNA samples for each SNV of *FGFR3* and *TERT* in the multiplex ddPCR assays.

Accuracy was determined as the percentage of the expected mutant DNA concentration compared with the mutant + WT DNA sample. Quantification of mutant DNA within the combined sample was then reviewed against its corresponding 95% CI.

#### 2.3.5. Analytical Precision

Analytical precision was defined as reproducibility within a single run or between separate runs for replicate samples. Variance in the ddPCR assay was assessed using HECs and LECs for each *FGFR3* and *TERT* SNV. A fitted linear random effects model was used to estimate intra-assay and inter-assay variance for the mutant fraction, presented as standard deviation (SD) and 95% CI; the coefficient of variation (CV%) was also calculated. Inter-assay, intra-assay, and total variance for the mutant fraction were assessed by four operators over >60 days by reviewing 46 ddPCR plate controls split over 22 plates (per *FGFR3* SNV) or 48 ddPCR plate controls over 23 plates (per *TERT* SNV).

The lot-to-lot reagent variation was also assessed by testing three independent manufacture lots of MagMAX magnetic beads (BD2206339, BD2206338, and BD2302342) and MagMAX wash buffer (WB2405057, WB2405056, and WB2312054) for both *FGFR3* and *TERT* SNVs using HECs and LECs. Each manufacture lot was run in parallel on a single plate with at least eight replicates of each control for each lot of reagents.

#### 2.3.6. Extraction Efficiency

To confirm that Triage Plus was not biased for extraction of either *FGFR3* or *TERT* mutant DNA, synthetic urine samples were prepared for each SNV at high (1:10) and low (1:200) mutant-to-WT ratios. The extraction efficiency was tested by processing eight replicates of each synthetic urine sample from extraction to ddPCR. These results were then compared with the expected mutant fraction and copy number. The absolute quantity of the extracted sample was used to define the extraction efficiency of DNA.

#### 2.3.7. Inter-Laboratory Evaluation

The inter-laboratory comparison between the NZ laboratory (PEDNZ) and the US laboratory (PEDUSA) was assessed for Cxbladder Triage Plus. A random sample set from the PEDNZ validation was used to confirm the reproducibility of Triage Plus at PEDUSA. Acceptable variability was defined as achieving ≥80% concordance for all clinical results.

### 2.4. Statistical Analysis

Statistical analyses were conducted using R, version 4.3.1.

## 3. Results

### 3.1. Algorithm Development Samples

Of the original 1073 samples, 63 had DNA results that were neither positive nor negative and 23 had data missing for at least one of the predictors (*IGFBP5*, *HOXA13*, *MDK*, *CDK1*, and *CXCR2*) and could not be used to fit the Triage Plus algorithm. Therefore, 987 samples were used for the development of Triage Plus (Appendix A).

### 3.2. Predicted Performance

Using this development dataset, the performance of the Triage Plus algorithm (score threshold 0.15) gave a sensitivity of 93.6%, a specificity of 90.8%, a PPV of 46.5%, an NPV of 99.4%, and a TNR of 84.1%. Using the upper score threshold (0.54), Triage Plus had higher specificity (98.2%) and PPV (74.6%). Confusion matrices for the predicted performance Triage Plus algorithm versus tumor status confirmed by pathology are shown in Appendix A.

When the lower and upper score thresholds for Triage Plus were used to define the probability of UC, the actual incidence of UC (confirmed by pathology) was 0.6% in low-probability samples (score < 0.15; 84.1% of samples), 27.7% in intermediate-probability samples (score ≥ 0.15 to <0.54; 9.5% of samples), and 74.6% in high-probability samples (score ≥ 0.54; 6.4% of samples).

### 3.3. Analytical Validation

The analytical validation of Triage Plus for detection of mRNA from *IGFBP5*, *HOXA13*, *MDK*, *CDK1*, and *CXCR2* genes was the same as previously described for Cxbladder Detect and Triage [16]. The analytical validation of Triage Plus for detection of mutant DNA SNVs from *FGFR3* and *TERT* is described in the following sections.

#### 3.3.1. Linearity

Triage Plus demonstrated linearity across all analyzed *FGFR3* and *TERT* SNVs, with an *R*^2^ value of >0.99 for all targets (Figure 1). The MSE values for individual SNVs ranged from 0.006 to 0.014. The tested range was 1 × 10^5^ to 1 × 10^1^ DNA copies/well, which corresponded to a maximum linear concentration of 5λ. The linearity of the assay was considered acceptable as it met the criteria for linearity in diagnostic testing.

#### 3.3.2. Analytical Sensitivity

The predicted LOD of Triage Plus for DNA detection using the logistic regression approach was a mutant-to-WT DNA ratio of 1:840, 1:1200, 1:1250, and 1:970 for *FGFR3* R248C, S249F/C, G372C, and Y375C, respectively, and a mutant-to-WT DNA ratio of 1:440 and 1:740 for *TERT* C228T and C250T, respectively (Table 1). Using the three concentrations approach, the predicted LOD was a mutant-to-WT DNA ratio of 1:632, 1:1220, 1:946, and 1:439 for *FGFR3* R248C, S249F/C, G372C, and Y375C, respectively, and a mutant-to-WT DNA ratio of 1:319 and 1:418 for *TERT* C228T and C250T, respectively. There was no significant difference between the logistic regression and three concentrations approach for LOD prediction; however, the logistic regression approach was preferred for all SNVs except *FGFR3* G372C, for which the logistic regression approach did not work well as there were too few negative samples.

#### 3.3.3. Analytical Specificity

For sample-derived substances, RBCs had no significant effect on the extraction of HECs or LECs of *FGFR3* and *TERT* at RBC concentrations below 4 × 10^6^ cells/mL (Table 2). At RBC concentrations of 2 × 10^7^ cells/mL, the *FGFR3* and *TERT* mutant count per well was increased for HECs and decreased for LECs. At these RBC concentrations, all controls returned a positive Triage Plus result. At RBC concentrations of 1 × 10^8^ cells/mL, there was significant impact on extraction efficiency; however, there was no loss of sensitivity as all HECs returned a positive Triage Plus result. LECs had some loss of sensitivity at RBC concentrations of 1 × 10^8^ cells/mL, with the potential to return a false-negative test result.

The presence of clinically high levels of bacteria (*E. coli* 1 × 10^6^ cells/mL), glucose (0.5 mg/mL), or yeast (1 × 10^4^ CFU/mL) had no significant effect on the extraction of HECs and LECs of *FGFR3* and *TERT* or assay performance. However, the presence of urea (60 mg/mL) had an impact on the extraction efficiency of HEC and increased the *FGFR3* mutant count per well, but did not impact Triage Plus results. The presence of protein had an effect on the extraction efficiency and DNA detection, with the results for 1.25 mg/mL serum albumin showing a significant (but not meaningful) effect on extraction efficiency. For HECs, a positive result was returned at any protein (serum albumin) concentration; however, LECs were not guaranteed to return a positive result at serum albumin concentrations of ≥2.5 mg/mL. The required minimum WT count was not reached for samples containing ≥2.5 mg/mL serum albumin. For process-derived substances, all reagents showed a slight increase in assay performance, but were within the expected level of variation.

The substances that had the greatest impact on the assay (MagMAX wash buffer and Cxbladder stabilizing reagent) are the first and second reagents used in the extraction and purification steps, and were considered to have a low risk of being present at high concentrations at the elution step (Appendix A). The substances that are closest to the elution step (acetone and MagMAX magnetic beads) did not significantly impact the assay at the highest concentrations tested.

#### 3.3.4. Analytical Accuracy

The absolute quantification of *FGFR3* mutant DNA versus mutant + WT DNA were within the 95% CI for the expected intra-plate variance for HECs of R248C and S249F/C and LECs of S249F/C and G372C (Table 3). However, higher intra-plate variance was observed for the LEC of R248C, HEC of G372C, and the HEC and LEC of Y373.

*TERT* mutant DNA versus mutant + WT DNA quantification showed acceptable intra-plate variance for HECs of C228T and C228T + C250T, but higher acceptable intra-plate variance for all LECs and the HEC of C250T.

#### 3.3.5. Analytical Precision

Inter-assay variability showed mutant fraction CV% of ≤4.33% for HECs, with mutant fraction CV%s of 1.35–4.33% for *FGFR3* and 1.01–2.11% for *TERT*; however, mutant fraction variance was slightly higher for LECs (mutant fraction CV%s of 4.62–6.97% and 3.42–6.86%, respectively; Table 4). Based on a maximum mutant fraction CV% of 6.97% for *FGFR3* C228T and 6.86% for *TERT* C228T + C250T, the inter-assay variability was deemed acceptable.

For intra-assay variability, HECs had mutant fraction CV%s of ≤3.54% for *FGFR3* and ≤2.95% for *TERT* (Table 4). LECs had higher mutant fraction CV%s than HECs (11.56–15.90% for *FGFR3* and 11.79–20.27% for *TERT*). Based on a maximum mutant fraction CV% of 15.90% for *FGFR3* G372C and 20.27% for *TERT* C228T + C250T, the intra-assay variability was deemed acceptable.

Total assay mutant fraction CV% for *FGFR3* and *TERT* was ≤5.59% for HECs, and 12.45–17.24% and 12.27–21.40%, respectively, for LECs (Table 4). The maximum total mutant fraction variance was considered excellent for high-extraction controls and acceptable for low-extraction controls.

Lot-to-lot reagent variance was low for *FGFR3* HECs (mutant fraction CV%s 1.14–2.53%), and within an acceptable range (CV%s 13.10–17.64%) for LECs (Appendix A). Similarly, lot-to-lot variance was low for *TERT* HECs (mutant fractions CV%s 0.84–1.74%), but was higher for LECs (CV%s 11.46–32.15%).

#### 3.3.6. Extraction Efficiency

The extraction efficiency of Triage Plus for *FGFR3* and *TERT* SNVs was lower in HECs versus LECs (Table 5). The *FGFR3* samples had a mean extraction efficiency of 72.9% for high mutant-to-WT ratio controls and 88.0% for low mutant-to-WT ratio controls. The *TERT* samples had a mean extraction efficiency of 83.5% for high mutant-to-WT ratio controls and 95.5% for low mutant-to-WT ratio controls. No difference was observed between extracted samples and input controls, confirming that the extraction method did not introduce sampling bias.

#### 3.3.7. Inter-Laboratory Evaluation

The inter-laboratory comparison between PEDNZ and PEDUSA was based on data from 33 samples. There was 87.9% concordance in clinical results for Triage Plus between the two laboratories using assay-derived CIs as the concordance metric.

## 4. Discussion

This analytical validation study showed that Cxbladder Triage Plus can accurately and reproducibly quantify the presence of six DNA SNVs of *FGFR3* and *TERT*, as well as mRNA expression of five biomarker genes, to provide risk stratification for UC in patients with hematuria. All pre-specified analytical criteria were met, including analytical linearity, sensitivity, specificity, accuracy, and precision. The most notable impact on specificity was observed with protein contamination, which showed an effect on the accuracy of the quantitative ddPCR assay, but did not meaningfully change the ability to detect the presence of DNA mutations at concentrations of <2.5 mg/mL.

The original Cxbladder Triage assay was developed to provide high sensitivity and NPV when risk stratifying patients with hematuria for possible UC, while Cxbladder Detect was designed for high specificity [6,7]. The predicted performance of Triage Plus showed sensitivity and NPV that was as good as, or better than, Cxbladder Triage, as well as significantly higher specificity and PPV. Triage Plus also had improved performance over Cxbladder Detect. Furthermore, clinical validation of Triage Plus found that this assay provides improved specificity versus Cxbladder Triage and improved sensitivity versus Cxbladder Detect in a Veterans Affairs population with hematuria [19].

In the current study, Triage Plus showed acceptable accuracy for measurement of high and low controls containing *FGFR3* and *TERT* mutant DNA. Analytical precision analysis showed acceptable inter-assay, intra-assay, and total assay variance, although variance was greater in low controls than in high controls. A study by Liu and colleagues similarly showed increased variance as the mutant frequency decreased [20]. Lot-to-lot reagent variance was low for HECs and acceptable for LECs. The analytical validity of Triage Plus was confirmed at a second laboratory (PEDUSA), with concordance between the two laboratories of 87.9%, consistent with that reported for Cxbladder Triage and Detect [16].

Of note, the presence of protein was found to have an impact on analytical specificity of Triage Plus and ddPCR-based DNA detection. While a DNA-positive call was returned at all protein concentrations for HECs, this could not be confidently asserted for LECs containing protein concentrations of ≥2.5 mg/mL. Although this is considered to be a very high urine protein concentration, patients with severe proteinuria can have protein concentrations as high as 30 mg/mL. The inhibitory effect of protein was observed for all DNA SNVs and resulted in test failure for WT only samples. For all samples that were impacted by protein contamination (i.e., patients with proteinuria), positive samples would return either a positive or “No Result” finding for Triage Plus. In clinical practice, the protein-inhibited result would not affect patient outcomes, as a “No Result” finding would require further evaluation for UC as part of standard of care.

The limitations of this study include its analytical validation design, which means that it does not provide clinical validation or describe the clinical utility of this assay. However, data from the recent DRIVE study demonstrated that Triage Plus has clinical validity for the risk stratification of patients presenting with hematuria [19].

## 5. Conclusions

This analytical validation study demonstrated that the urinary biomarker Cxbladder Triage Plus assay can accurately and reproducibly detect six DNA SNVs of *FGFR3* and *TERT*, as well as mRNA expression of five biomarker genes that are associated with UC, from urine samples of patients with hematuria. This assay will provide clinicians with a non-invasive method of risk stratification in patients presenting for evaluation of hematuria, thereby allowing for more accurate assessment of UC risk in these patients.

## Figures and Tables

**Figure 1 diagnostics-15-01739-f001:**
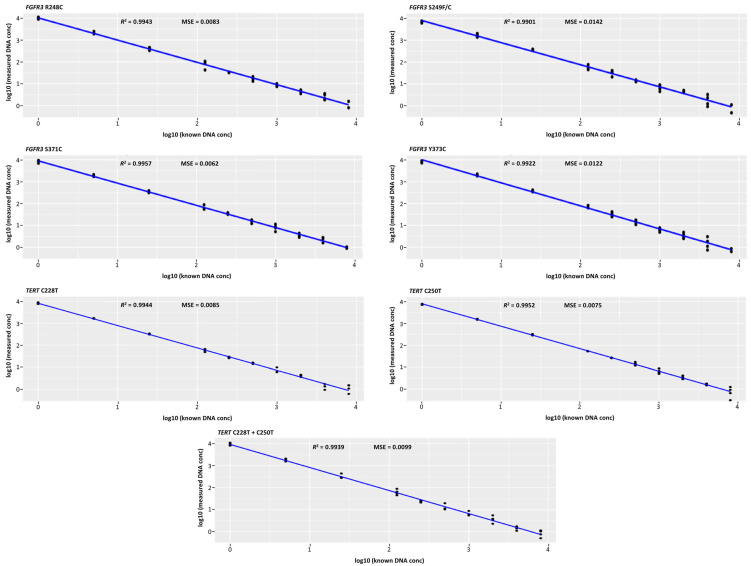
Measured versus known DNA concentrations for the six single-nucleotide variants of *FGFR3* and *TERT*. conc, concentration; *FGFR3*, fibroblast growth factor receptor 3; MSE, mean squared error; *R*^2^, regression coefficient; *TERT*, telomerase reverse transcriptase.

**Table 1 diagnostics-15-01739-t001:** Limit of detection (analytical sensitivity) of Cxbladder Triage Plus for the six DNA single nucleotide variants of *FGFR3* and *TERT*.

SNV	Estimated LOD (95% CI)	
	Logistic Regression	Three Concentrations
*FGFR3*, mutant-to-WT DNA ratio		
R248C	1:840 (1:600, 1:980)	1:632 (1:468, 1:978)
S249F/C	1:1200 (1:860, 1:1370)	1:1220 (1:806, 1:2779)
G372C	1:1250 ^a^	1:946 (1:630, 1:2263)
Y375C	1:970 (NE ^b^, 1:1220)	1:439 (1:286, 1:2128)
*TERT*, mutant-to-WT DNA ratio		
C228T	1:440 (1:250, 1:520)	1:319 (1:217, 1:1494)
C250T	1:740 (1:560, 1:810)	1:418 (1:300, 1:651)

^a^ The logistic regression approach did not work well for *FGFR3* G372C as there were too few negative samples. ^b^ The lower 95% CI was NE due to large variations in the data. CI, confidence interval; *FGFR3*, fibroblast growth factor receptor 3; LOD, limit of detection; NE, not estimable; SNV, single-nucleotide variant; *TERT*, telomerase reverse transcriptase; WT, wild type.

**Table 2 diagnostics-15-01739-t002:** Analytical specificity of Cxbladder Triage Plus for *FGFR3* and *TERT* DNA control samples when mixed with sample-derived interfering substances.

Substance	*FGFR3* Mutant Count per Well			*TERT* Mutant Count per Well		
	Mean	Difference ^a^	*p*-Value ^b^	Mean	Difference ^a^	*p*-Value ^b^
RBCs, cells/mL						
HECs	767.60	–	–	692.14	–	–
8 × 10^5^	763.13	−4.48	0.894	729.00	+36.86	0.120
4 × 10^6^	781.38	+13.78	0.694	733.63	+41.48	0.120
2 × 10^7^	847.00	+79.40	0.049	811.25	+119.11	0.000
1 × 10^8^	535.63	−231.98	0.000	468.50	−223.64	0.000
LECs	33.00	–	–	31.71	–	–
8 × 10^5^	31.43	−1.57	0.730	30.50	−1.21	0.808
4 × 10^6^	24.33	−8.67	0.074	26.00	−5.71	0.043
2 × 10^7^	23.86	−9.14	0.053	20.50	−11.21	0.003
1 × 10^8^	6.13	−26.88	0.000	3.17	−28.55	0.000
Bacteria (*E. coli*), cells/mL						
HECs	662.20	–	–	691.38	–	–
1 × 10^6^	716.63	+54.43	0.063	694.43	+3.05	0.834
LECs	14.25	–	–	12.71	–	–
1 × 10^6^	14.25	0.00	1.000	10.63	−2.09	0.205
Yeast, CFU/mL						
HECs	662.20	–	–	691.38	–	–
1 × 10^4^	696.63	+34.43	0.064	667.00	−24.38	0.157
LECs	14.25	–	–	12.71	–	–
1 × 10^4^	14.14	−0.11	0.935	9.38	−3.34	0.020
Urea, mg/mL						
HECs	662.20	–	–	691.38	–	–
60	736.75	+74.55	0.010	701.43	+10.05	0.470
LECs	14.25	–	–	12.71	–	–
60	14.63	+0.38	0.813	13.50	+0.79	0.638
Glucose, mg/mL						
HECs	662.20	–	–	691.38	–	–
0.5	756.57	+94.37	0.000	706.71	+15.34	0.365
LECs	14.25	–	–	12.71	–	–
0.5	17.38	+3.13	0.169	14.71	+2.00	0.217
Protein (serum albumin), mg/mL						
HECs	683.50	–	–	754.88	–	–
1.25	594.25	−89.25	0.000	630.13	−124.75	0.000
2.50	8.88	−674.63	0.000	7.63	−747.25	0.000
5.00	6.29	−677.21	0.000	3.88	−751.00	0.000
10.00	4.29	−679.21	0.000	4.63	−750.25	0.000
LECs	15.00	–	–	10.83	–	–
1.25	11.75	−3.25	0.256	13.13	+2.29	0.174
2.50	0.00	−15.00	0.002	0.00	−10.83	0.000
5.00	0.00	−15.00	0.002	0.00	−10.83	0.000
10.00	0.00	−15.00	0.002	0.00	−10.83	0.000

^a^ Calculated by subtracting the mean score for the control sample from that of each contaminated sample. ^b^ Two-sided t test. CFU, colony-forming unit; *FGFR3*, fibroblast growth factor receptor 3; HECs, high-extraction controls; LECs, low-extraction controls; RBC, red blood cell; *TERT*, telomerase reverse transcriptase.

**Table 3 diagnostics-15-01739-t003:** Analytical accuracy of Cxbladder Triage Plus for the six DNA single-nucleotide variants of *FGFR3* and *TERT*.

SNV	DNA Concentration, Copies/μL		Difference	Inaccuracy (95% CI), %
	Mutant DNA	Mutant + WT DNA		
*FGFR3*				
R248C				
HEC ^a^	237.69	239.08	1.38	0.6 (−11.8, 13.0)
LEC ^b^	11.16	14.38	3.22	28.9 (14.5, 43.3)
S249F/C				
HEC ^a^	193.48	201.64	8.17	4.2 (−6.7, 15.2)
LEC ^b^	10.18	10.50	0.32	3.1 (−11.0, 17.3)
G372C				
HEC ^a^	197.82	225.15	27.34	13.8 (3.6, 24.0)
LEC ^b^	11.25	10.09	−1.16	−10.3 (−22.4, 1.8)
Y375C				
HEC ^a^	232.29	247.06	14.77	6.4 (1.3, 11.5)
LEC ^b^	8.76	12.46	3.70	42.2 (18.2, 66.3)
*TERT*				
C228T				
HEC ^a^	229.43	237.82	8.39	3.7 (−2.5, 9.9)
LEC ^b^	10.41	13.43	3.02	29.0 (14.5, 43.4)
C250T				
HEC ^a^	187.32	216.35	29.03	15.5 (9.5, 21.5)
LEC ^b^	9.08	11.81	2.74	30.2 (19.9, 40.4)
C228T + C250T				
HEC ^a^	226.62	240.62	14.0	6.2 (−12.8, 25.1)
LEC ^b^	9.41	15.16	5.75	61.1 (28.7, 93.5)

^a^ DNA concentration of ~1 × 10^6^ copies/μL and mutant-to-WT ratio of 1:10 for combined sample. ^b^ DNA concentration of ~1 × 10^4^ copies/μL and mutant-to-WT ratio of 1:200 for combined sample. CI, confidence interval; *FGFR3*, fibroblast growth factor receptor 3; HEC, high-extraction control; LEC, low-extraction control; SNV, single-nucleotide variant; *TERT*, telomerase reverse transcriptase; WT, wild type.

**Table 4 diagnostics-15-01739-t004:** Analytical precision of Cxbladder Triage Plus for the six DNA single-nucleotide variants of *FGFR3* and *TERT*.

SNV	Mutant Fraction Variance					
	Inter-Assay Variance		Intra-Assay Variance		Total Assay Variance	
	Mean (SD)	CV%	Mean (SD)	CV%	Mean (SD)	CV%
*FGFR3*						
R248C						
HEC ^a^	0.1047 (0.0014)	1.35	0.1100 (0.0025)	2.39	0.1047 (0.0029)	2.74
LEC ^b^	0.0063 (0.0003)	4.64	0.0063 (0.0008)	13.22	0.0063 (0.0009)	14.01
S249F/C						
HEC ^a^	0.0808 (0.0035)	4.33	0.0808 (0.0029)	3.54	0.0808 (0.0045)	5.59
LEC ^b^	0.0043 (0.0002)	4.62	0.0043 (0.0005)	11.56	0.0043 (0.0005)	12.45
G372C						
HEC ^a^	0.0909 (0.0014)	1.50	0.0909 (0.0029)	3.15	0.0909 (0.0032)	3.49
LEC ^b^	0.0053 (0.0004)	6.68	0.0053 (0.0008)	15.90	0.0053 (0.0009)	17.24
Y375C						
HEC ^a^	0.0965 (0.0028)	2.89	0.0964 (0.0028)	2.90	0.0964 (0.0040)	4.10
LEC ^b^	0.0058 (0.0004)	6.97	0.0057 (0.0009)	15.02	0.0057 (0.0009)	16.56
*TERT*						
C228T						
HEC ^a^	0.1033 (0.0014)	1.38	0.1033 (0.0025)	2.45	0.1033 (0.0029)	2.82
LEC ^b^	0.0063 (0.0004)	6.18	0.0063 (0.0008)	12.42	0.0063 (0.0009)	13.87
C250T						
HEC ^a^	0.0913 (0.0019)	2.11	0.0913 (0.0023)	2.54	0.0913 (0.0030)	3.30
LEC ^b^	0.0055 (0.0002)	3.42	0.0055 (0.0007)	11.79	0.0055 (0.0007)	12.27
C228T + C250T						
HEC ^a^	0.0933 (0.0009)	1.01	0.0933 (0.0027)	2.95	0.0933 (0.0029)	3.11
LEC ^b^	0.0054 (0.0004)	6.86	0.0054 (0.0011)	20.27	0.0054 (0.0012)	21.40

^a^ DNA concentration of ~1 × 10^6^ copies/μL and mutant-to-WT ratio of 1:10. ^b^ DNA concentration of ~1 × 10^4^ copies/μL and mutant-to-WT ratio of 1:200. CV%, coefficient of variation; *FGFR3*, fibroblast growth factor receptor 3; HEC, high-extraction control; LEC, low-extraction control; SD, standard deviation; SNV, single-nucleotide variant; *TERT*, telomerase reverse transcriptase; WT, wild type.

**Table 5 diagnostics-15-01739-t005:** Extraction efficiency of Cxbladder Triage Plus for the six DNA single-nucleotide variants of *FGFR3* and *TERT*.

SNV	Extraction Efficiency (95% CI), %
*FGFR3*	
R248C	
HEC ^a^	72.7 (69.3, 76.0)
LEC ^b^	82.4 (51.4, 100.0)
S249F/C	
HEC ^a^	62.9 (53.3, 72.5)
LEC ^b^	84.2 (59.6, 100.0)
G372C	
HEC ^a^	77.6 (69.5, 85.7)
LEC ^b^	100.0 (100.0, 100.0)
Y375C	
HEC ^a^	78.6 (70.1, 87.2)
LEC ^b^	85.6 (60.7, 100.0)
*TERT*	
C228T	
HEC ^a^	86.4 (76.1, 96.6)
LEC ^b^	97.5 (71.1, 100.0)
C250T	
HEC ^a^	83.4 (76.7, 90.1)
LEC ^b^	95.2 (69.8, 100.0)
C228T + C250T	
HEC ^a^	80.7 (70.5, 90.9)
LEC ^b^	93.8 (52.6, 100.0)

^a^ DNA concentration of ~1 × 10^6^ copies/μL and mutant-to-WT ratio of 1:10. ^b^ DNA concentration of ~1 × 10^4^ copies/μL and mutant-to-WT ratio of 1:200. CI, confidence interval; *FGFR3*, fibroblast growth factor receptor 3; SNV, single-nucleotide variant; *TERT*, telomerase reverse transcriptase; WT, wild type.

## Data Availability

The raw data supporting the conclusions of this article will be made available by the authors upon reasonable request.

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
