# Peer review of "Analytical Validation of the Cxbladder® Triage Plus Assay for Risk Stratification of Hematuria Patients for Urothelial Carcinoma"

_diagnostics, 2025, doi:10.3390/diagnostics15141739_

Round 1
Reviewer 1 Report
Comments and Suggestions for Authors
This study provides a breakthrough tool for risk assessment of urothelial carcinoma (UC) in hematuria patients through technological innovation, rigorous validation, and clinical orientation. Its core advantages lie in the integrated strategy of multimodal biomarkers and the high-precision ddPCR detection technology, combining scientific value with clinical translation potential. Although there are limitations such as insufficient clinical validation, its methodological breakthroughs and detailed data still provide important references for the field of non-invasive cancer diagnosis.
Comments on the Quality of English LanguageThis study provides a breakthrough tool for risk assessment of urothelial carcinoma (UC) in hematuria patients through technological innovation, rigorous validation, and clinical orientation. Its core advantages lie in the integrated strategy of multimodal biomarkers and the high-precision ddPCR detection technology, combining scientific value with clinical translation potential. Although there are limitations such as insufficient clinical validation, its methodological breakthroughs and detailed data still provide important references for the field of non-invasive cancer diagnosis.
Author Response
Comment 1: This study provides a breakthrough tool for risk assessment of urothelial carcinoma (UC) in hematuria patients through technological innovation, rigorous validation, and clinical orientation. Its core advantages lie in the integrated strategy of multimodal biomarkers and the high-precision ddPCR detection technology, combining scientific value with clinical translation potential. Although there are limitations such as insufficient clinical validation, its methodological breakthroughs and detailed data still provide important references for the field of non-invasive cancer diagnosis.
Response 1: Thank you for your positive comment. No changes to the manuscript have been made in response to this comment.
Comment 2: The English could be improved to more clearly express the research.
Response 1: The manuscript has been reviewed and edited by a professional medical writer to improve the quality of the English.
Reviewer 2 Report
Comments and Suggestions for Authors
Dear Author
The following are my comments:
- The logic of choosing six SNPs from two target genes is not clear “ six DNA single-nucleotide polymorphisms (SNPs) from two genes (fibroblast growth factor receptor 3 [FGFR3] and telomerase reverse transcriptase [TERT]) targets to provide risk stratification for urothelial carcinoma (UC) in patients with hematuria”. Are they frame shift SNPs or nonsynonymous amino acid change?
- In Algorithm Development part In the original Detect+ algorithm, the mRNA results were encapsulated into two variables for the algorithm: X1, which combined the four cancer-related biomarkers (IGFBP5, HOXA13, MDK, and CDK1) into a single predictor; and X2, the inflammation target (CXCR2). The DNA component of the original Detect+ algorithm was summarized as a “FGFR3 and TERT DNA-positive calling”. Should be explained more. The role of IGFBP5, HOXA13, MDK, and CDK1 in algorithm X1 and its relation to “FGFR3 and TERT DNA-positive calling
- The analytical validation of Triage Plus for detection of mRNA from IGFBP5 HOXA13, MDK, CDK1, and CXCR2 genes was the same as previously described for Cxbladder Detect and Triage [13]. The reference 13 is previous work of Dr Justin C. Harvey “Analytical validation of Cxbladder® Detect, Triage, and Monitor: assays for detection and management of urothelial carcinoma”. Sane as reference 7 (self-citations)
Author Response
Comment 1: The logic of choosing six SNPs from two target genes is not clear “ six DNA single-nucleotide polymorphisms (SNPs) from two genes (fibroblast growth factor receptor 3 [FGFR3] and telomerase reverse transcriptase [TERT]) targets to provide risk stratification for urothelial carcinoma (UC) in patients with hematuria”. Are they frame shift SNPs or nonsynonymous amino acid change?
Response 1: Previous analyses have demonstrated that hotspot mutations in the FGFR3 gene (R248, S249C, G372C and Y375C [missense gain-of-function mutations]) and upstream of the TERT promoter (g.1295228 C>T and g.1295250 C>T [gain-of-function mutations]) are often identified in urothelial carcinoma samples (Pietzak EJ, et al. Eur Urol 2017, 72, 952–959; Hayashi Y, et al. Front Oncol 2020, 10, 755; Knowles MA. Bladder Cancer 2020, 6, 403–423; now refs #11–13). This information has been added to the Introduction of the manuscript (paragraph 4). These findings provided the rationale for the inclusion of the six single-nucleotide variant (SNV) mutation targets in Cxbladder Triage Plus.
Comment 2: In Algorithm Development part in the original Detect+ algorithm, the mRNA results were encapsulated into two variables for the algorithm: X1, which combined the four cancer-related biomarkers (IGFBP5, HOXA13, MDK, and CDK1) into a single predictor; and X2, the inflammation target (CXCR2). The DNA component of the original Detect+ algorithm was summarized as a “FGFR3 and TERT DNA-positive calling”. Should be explained more. The role of IGFBP5, HOXA13, MDK, and CDK1 in algorithm X1 and its relation to “FGFR3 and TERT DNA-positive calling.”
Response 2: The “FGFR3 and TERT DNA-positive calling” refers to identification of FGFR3 and TERT SNV DNA in the sample (resulting in a positive DNA call). In the original algorithm, the X1 variable in the algorithm represented quantification of mRNA expression of the four biomarkers known to be associated with UC (IGFBP5, HOXA13, MDK, and CDK1; Holyoake, A. et al. Clin. Cancer Res. 2009, 14, 742-749; ref #8), while X2 represented quantification of mRNA expression of the inflammation biomarker (CXCR2). The updated algorithm had a separate variable for each biomarker (X1 = IGFBP5, X2 = HOXA13, X3 = MDK, X4 = CDK1, and X5 = CXCR2), appearing separately in a second-order polynomial equation, with coefficients (a0, a1, …, a45) obtained by fitting a logistic regression model with the confirmed diagnosis used as the response variable. We have revised the text in the Methods section of the manuscript (2.2; paragraphs 1, 3, and 4) to clarify these points.
Comment 3: The analytical validation of Triage Plus for detection of mRNA from IGFBP5 HOXA13, MDK, CDK1, and CXCR2 genes was the same as previously described for Cxbladder Detect and Triage [13]. The reference 13 is previous work of Dr Justin C. Harvey “Analytical validation of Cxbladder® Detect, Triage, and Monitor: assays for detection and management of urothelial carcinoma”. Sane as reference 7 (self-citations).
Response 3: This citation refers to an earlier publication from our research group, in which we reported the analytical validation of the original Cxbladder Detect and Triage assays. As the analytical validity of these assays for detection of mRNA for these biomarker genes had been previously established and was not changed in Cxbladder Triage Plus, the analytical performance of mRNA detection in Triage Plus was only referred to in our manuscript (not reiterated). The current study focused on demonstrating the analytical validity of Triage Plus for detection of DNA SNP mutations from the FGFR3 and TERT genes.